# Metabolomic Approach in STEMI-Patients Undergoing Left Ventricular Remodeling

**DOI:** 10.3390/ijms20020289

**Published:** 2019-01-12

**Authors:** Gabriel Garcia, Juan Manuel Chao de la Barca, Delphine Mirebeau-Prunier, Pascal Reynier, Alain Furber, Fabrice Prunier, Loïc Bière

**Affiliations:** 1Institut Mitovasc, University of Angers, University Hospital of Angers, 49000 Angers, France; garcia.gabriel.85@gmail.com (G.G.); JMChaoDeLaBarca@chu-angers.fr (J.M.C.d.l.B.); DePrunier@chu-angers.fr (D.M.-P.); pareynier@chu-angers.fr (P.R.); AlFurber@chu-angers.fr (A.F.); FaPrunier@chu-angers.fr (F.P.); 2UMR CNRS 6015-INSERMU1083, 49000 Angers, France; 3Department of Cardiology, University Hospital of Angers, 49000 Angers, France; 4Department of Biochemistry, University Hospital of Angers, 49000 Angers, France

**Keywords:** left ventricular remodeling, myocardial infarction, metabolomics, cardiac magnetic resonance

## Abstract

Left ventricular remodeling (LVR) occurring after ST-segment elevation myocardial infarction (STEMI) is frequent and severe. We present a metabolomic approach as an attempt to reveal unknown biomarkers associated with post-STEMI LVR. Out of 192 consecutive patients with successfully revascularized STEMI, 32 presented LVR and were clinically matched with 32 no-LVR patients. They underwent cardiac magnetic resonance at baseline, three months and 12 months. Blood samples were collected during index hospitalization. Creatine kinase (CK) peak and inflammatory markers were higher for LVR patients compared to no-LVR patients (mean 3466 ± 2211 and 2394 ± 1615 UI/L respectively, *p* = 0.005 for CK peak; mean 35.9 ± 44.3 vs. 21.7 ± 30.4 mg/L respectively, *p* = 0.020 for C-reactive protein). Leukocyte and neutrophil counts were also higher for LVR patients (mean 12028 ± 2593/mL vs. 10346 ± 3626/mL respectively, *p* = 0.028 and mean 9035 ± 3036/mL vs. 7596 ± 3822/mL respectively, *p* < 0.001). For metabolomic analysis, sphingomyelin C20:2 and symmetrical dimethylarginine were higher for LVR patients, but did not reach significance after the correction for the alpha risk. The metabolomic approach did not discriminate patients with and without LVR. However, common parameters that focus on infarction severity, such as infarct size and inflammatory markers, differed between the groups.

## 1. Introduction

In recent decades, despite improvements in the management of myocardial infarction, left ventricular remodeling (LVR) has remained a severe and common issue, estimated to occur in up to 30% of cases following ST-segment elevation myocardial infarction (STEMI). LVR can lead to chronic heart failure, which is one of the primary causes of death worldwide [1]. Although some mechanisms are well described and many biomarkers are associated with LVR [2], none of them show consistent results, particularly in terms of prognosis, or in providing potential therapeutic targets.

Metabolomics can be defined as the detection, quantification and identification of low molecular weight molecules of the metabolism [3]. In comparison with other “-omics” approaches, such as genomics, transcriptomics and proteomics, metabolomics closely explores the phenotype and more accurately reflects the functioning of a biological system at a given time [4]. Recently, Cheng et al. showed that the diagnostic value of some metabolites encountered in heart failure, such as histidine or phenylalanine, was similar to brain natriuretic peptide (BNP), but with superior prognostic value [5]. We propose that a similar approach would be of interest in STEMI patients. Hence, the objective of this study was to identify new biomarkers associated with post-STEMI LVR using a metabolomic approach.

## 2. Results

### 2.1. Baseline Characteristics

Overall, patients were 58 ± 9 years old, 81% were male and 53% had anterior infarction. The mean left ventricular ejection fraction (LVEF) was 47.8 ± 9% and the mean infarct size was 16.1 ± 11.1% of the total left ventricle (LV). LV volumes were assessed by cardiac magnetic resonance (CMR). LVR occurred in 32 patients (16.7%) from a cohort of 192 STEMI patients. They were matched with 32 no-LVR patients for the purpose of the study. LVR patients presented more history of hypertension than no-LVR patients (56.3% vs. 25.0%, respectively, *p* = 0.01).

Mean change in LV end-systolic volume over time was −17.2 ± 1.3% among no LVR patients, and +22.2 ± 1.6% among LVR patients. Table 1 summarizes the main characteristics of the study population and Table 2 shows cardiac magnetic resonance (CMR) data.

### 2.2. Creatine Kinase Peak and Inflammatory Biomarkers

Creatine kinase (CK) peak was significantly higher among LVR patients than no-LVR patients (mean 3466 ± 2211 UI/L and 2394 ± 1615 UI/L respectively, *p* = 0.005) (Figure 1A). Similarly, inflammatory biomarkers were significantly higher among LVR patients: C-reactive protein (CRP) (35.9 ± 44.3 mg/L vs. 21.7 ± 30.4 mg/L, *p* = 0.020) (Figure 1B); leukocyte count (12028 ± 2593/mL vs. 10346 ± 3626/mL, *p* = 0.028) and neutrophil count (9035 ± 3036/mL vs. 7596 ± 3822/mL, *p* < 0.001) (Figure 1C). By allowing for the inflation of the alpha risk using Benjamini correction, the neutrophil count, CK peak and CRP remained significantly associated with LVR.

However, there was no association between LVR and N-terminal prohormone of brain natriuretic peptide (NT-proBNP) or creatinine (Table 3).

### 2.3. Metabolomic Analysis

After validation of the kit plate based on QC samples, 142 (75.5%) metabolites were kept for the statistical analysis: Free carnitine and 11 acylcarnitines, 21 amino acids, 12 biogenic amines, 12 lysophosphatidylcholines, 70 phosphatidylcholines, 14 sphingomyelins and the sum of hexoses.

In the univariate analysis, sphingomyelin C20:2 and symmetrical dimethylarginine (SDMA) were higher for the LVR group than for the no-LVR group (mean 0.29 ± 0.07 µmol/L vs. 0.25 ± 0.05 µmol/L respectively, *p* = 0.013 for sphingomyelin C20:2 and mean 0.57 ± 0.15 µmol/L vs. 0.52 ± 0.07 µmol/L respectively, *p* = 0.037 for SDMA), but did not reach significance after the correction of the alpha risk. Likewise, without reaching significance after Benjamini correction, lysophosphatidylcholine C17:0, phosphatidylcholine C40:6 and C42:3 were lower for the LVR group than for the no-LVR group (1.13 ± 0.30 µmol/L vs. 1.37 ± 0.46 µmol/L, *p* = 0.019 for lysophosphatidylcholine C17:0, 23.26 ± 6.08 µmol/L vs. 25.96 ± 6.48 µmol/L, *p* = 0.045 for phosphatidylcholine C40:6, and 0.47 ± 0.10 µmol/L vs. 0.52 ± 0.11 µmol/L, *p* = 0.037 for phosphatidylcholine C42:3). Paired principal component analysis (PCA) showed no outliers or grouping of patients according to their metabolic profile (Figure 2). Paired partial least squares discriminant analysis (PLS-DA) failed to discriminate between LVR and their paired controls, as global performance for predictive capabilities on the test sets was no different from the mean model (mean and median area under the Receiver Operating Characteristic (AUROC) equal to 0.48) (Figure 3). Only 11% of the AUROC were significantly different from 0.5, which is the predictive capability of the mean model. This result is in close agreement with a normal distributed AUROC of the null model where 5% of the AUROC models will, by chance, have significantly better predictive capabilities and another 5% will exhibit significantly worse predictive capabilities, with an overall percentage of significantly different models totaling 10%. Following our cut-off values for AUROC and its associate *p*-values, no variable selection was made. Univariate and multivariate unpaired tests showed similar results.

## 3. Discussion

STEMI patients with successful revascularization and optimal medical treatment showed higher creatine kinase peak and inflammatory biomarkers on admission. Nevertheless, metabolomics did not reveal a significant link between circulating metabolites and LVR.

Numerous biomarkers were of interest for predicting LVR [2]. These primarily include the reflection of cellular necrosis (creatine kinase, troponin) [6,7], wall stress (NT-proBNP) [8,9] and inflammation (CRP, leukocytes) [10,11]. These are therefore markers of myocardial infarction severity. In the present study, despite matching for infarct size and left ventricular ejection fraction, creatine kinase and biomarkers of inflammation were effectively higher in LVR patients (Table 3, Figure 1). To a lesser degree, biomarkers, such as glycemia [12], creatinine [13] or low density lipoprotein (LDL) cholesterol [14], have been linked with LVR, suggesting that the latter is a dynamic and multifactorial process, not caused exclusively by the severity of initial necrosis.

Using a proteomic approach, Fertin et al. revealed new avenues for exploring the physiopathology of LVR, thus demonstrating the involvement of the N-terminal human albumin fragment [15]. Here, we provide one of the first metabolomic approaches to studying LVR in STEMI patients. Recently, Cheng et al. have shown that several metabolites, including histidine, phenylalanine, spermidine and phosphatidylcholine C34:4, exhibited similar or even better diagnostic values for heart failure than BNP [5]. However, it is important to emphasize that a metabolomic signature was observed only for the most severely affected patients, which may present a limitation for the significance of these results and the interest in metabolomics for stable chronic disease.

There are potentially many metabolic pathways involved in the LVR phenomenon. Firstly, energy production necessary for the proper functioning of the heart is substantial. Multiple pathways involve numerous metabolites, some of which prove interesting for cardiovascular diseases, such as atherosclerosis [16] and myocardial infarction [17]. Heart failure is mostly studied through metabolomics, demonstrating a metabolic remodeling with a shift to glycolysis, which leads to important metabolomic disturbances, as reflected by the increase in circulating levels of lactate, pyruvate, non-esterified fatty acids and amino acids, such as valine or leucine [18]. Furthermore, in experimental studies of myocardial infarction, it was noted early on that metabolomic signals are characterized by major disturbances in tricarboxylic acid cycle and the accumulation of some amino acids, and metabolites are involved in the metabolism of pyrimidine and pentose phosphate pathways [17]. In our study, we were unable to identify specific metabolic pathways to distinguish between LVR and no-LVR patients soon after myocardial infarction.

Secondly, myocardial fibrosis is another major issue that occurs during LVR [19]. In experimental studies on a murine model, fibrosis was correlated with metabolites, such as trimethylamine N-oxide (TMAO) and choline [20]. However, these pathways presented delayed activation, seemingly after the fifth day, which might explain the lack of signal observed in our cohort.

Finally, endothelial metabolism could be affected. In fact, Lorin et al. showed that SDMA—an enzyme involved in oxidative stress by inhibiting the activity of the endothelial nitric oxide synthase—was a marker of left ventricular dysfunction and in-hospital mortality after a myocardial infarction [21]. In a cohort of 487 patients presenting with myocardial infarction, an increased SDMA level at baseline was associated with worse clinical outcomes through altered LVEF during hospital stays. Furthermore, SDMA was also associated with the prognosis of other cardiovascular diseases, such as atherosclerosis [22] and acute heart failure [23]. In our study, there was a significant difference for SDMA in univariate analysis (0.57 ± 0.15 µmol/L vs. 0.52 ± 0.07 µmol/L, respectively for LVR patients and no-LVR patients, *p* = 0.037), which was not confirmed after the risk alpha correction. There is some interest in continuing to assess the role of SDMA in larger cohorts.

Our study has several limitations: (1) The small sample size presents a limitation that we tried to constrain partially through the one-to-one matching; (2) Among the 40,000 metabolites currently known, our targeted analysis focused on only 188; and the analysis was performed at a single timepoint (48 h after STEMI onset); (3) The selected threshold for defining LVR, as an increase of 10% of the end-systolic volume, may seem low but is below interobserver variability in MRI [24] and our confirmative results for standard biomarkers verify our method; (4) For the matching, we did not take into consideration comorbidities, such as hypertension, diabetes or obesity, which may have their own metabolomic impact.

## 4. Material and Methods

### 4.1. Study Population

From January 2006 to December 2015, 192 patients presenting with their first STEMI, successfully managed within the first 12 h of the onset of chest pain by primary angioplasty with stent implantation, were included in a prospective cohort (French governmental hospital based clinical research program; PHRC, No.006/0070). The non-inclusion criteria were: Age < 18 years; history of myocardial infarction or coronary bypass grafting; cardiogenic shock on admission; severe comorbidities limiting life expectancy, and contraindications to CMR (pacemaker, metallic devices, claustrophobia or chronic renal insufficiency). CMR was performed at admission (day 6 [interquartile range (IQR) 4; 9]), at the 3-month (day 98 [IQR 94; 106]) and the 12-month follow-up (day 372 [IQR 368; 383]). This study was conducted in accordance with the principles of the Declaration of Helsinki and was approved by the hospital’s ethics committee (University Hospital of Angers, France). All patients gave written informed consent.

Clinical data including medical history, baseline characteristics and clinical presentation on admission were prospectively collected during the index hospitalization and at the 3 and 12-month follow-ups.

32 patients who presented LVR were matched with 32 no-LVR patients with regard to age, sex, left ventricular ejection fraction, infarct size, culprit artery and time to reperfusion.

### 4.2. CMR Data

#### 4.2.1. CMR Protocol

CMR was performed using a 3 Tesla image scan (Skyra 3T, Siemens Medical Solutions, Forchheim, Germany). Images were transferred to a workstation for analysis and calculation (QMass 7.1, Medis, Leiden, The Netherlands). Steady-state free precession sequences were used to produce cine-CMR images for multiple short-axis and long-axis two, three and four-chamber views. To cover the entirety of the left ventricle, contiguous 8mm-thick slices were required. First-pass perfusion images were obtained during the injection of gadolinium contrast agents at a dose of 0.2 mmol/kg (gadoterate meglumine, Dotarem^®^, Laboratoires Guerbet, Roissy-Charles de Gaulle, France) and after a recuperation-inversion sequence. Late gadolinium enhancement images were acquired for short-axis and long-axis four-chamber views, 12 min after the injection. In an isolated sequence, optimal inversion time for annulling the normal myocardial signal was determined. The parameters applied for the sequences were: Repetition time = 4.9 ms, excitation time = 1.9 ms, flip angle = 15°, slice thickness = 8 mm, and spatial resolution = 1.35 × 1.35 × 8 mm^3^.

#### 4.2.2. Image Analysis

Left ventricular end-systolic volume (LVESV), left ventricular end-diastolic volume (LVEDV), left ventricular ejection fraction and infarct size were calculated using cine-CMR images. Endocardial and epicardial borders were outlined manually on end-diastolic and end-systolic frames, excluding papillary muscles and the trabeculae.

Infarct size was quantified on late gadolinium enhancement imaging using the “full-width at half maximum” method.

LVR was defined as an increase of at least 10% of LVESV during the follow-up [25].

### 4.3. Biomarkers

Blood samples were obtained during the index hospitalization (48 h [IQR 24; 72]) and were stored at −80 °C.

#### 4.3.1. Metabolomics

Targeted quantitative metabolomic analyses were carried out using the Biocrates^®^ Absolute IDQ p180 kit (Biocrates Life sciences AG, Innsbruck, Austria). This kit, combined with a QTRAP 5500 mass spectrometer (SCIEX, Villebon-sur-Yvette, France), enabled quantification of up to 188 different metabolites of 6 different biochemical families. Their essential functions are structural and metabolic, and some of them are involved in signaling pathways. The full list of individual metabolites is available at http://www.biocrates.com/products/research-products/absoluteidq-p180-kit.

All reagents used in this analysis were of liquid chromatography mass spectrometry (LC-MS) grade and purchased from VWR (Fontenay-sous-Bois, France) and Merck (Molsheim, France). Sample preparation was performed following the Kit User Manual. Briefly, 10 microlitres of serum was extracted in a methanol solution using ammonium acetate after drying under nitrogen flow and derivatizing with phenyl isothiocyanate for the quantitation of amino acids and biogenic amines. The extracts were finally diluted with MS running solvent before flow injection analysis (FIA) and LC-MS/MS analysis. One blank sample (with no internal standard or sample added), three water-based zero samples (internal standards added and phosphate buffered saline as a “zero” sample), and three quality control samples were also added to the plate of the kit. A seven-point serial dilution of calibrators was added to the plate of the kit to generate calibration curves for the quantification of amino acids and biogenic amines.

Before statistical analysis, the raw data were examined in order to eliminate metabolites not measured accurately enough, i.e., metabolites with a concentration below the lower limit of quantitation (LLOQ) or above the upper limit of quantitation (ULOQ). When more than 20% of the concentration values were not measured accurately enough, the metabolite was not considered for further statistical analysis.

#### 4.3.2. Other Biomarkers

Routine biomarkers, analysed using routine automats (ARCHITECT c16000, Abbott, Chicago, USA) were: NT-proBNP (Immunochemiluminescence Roche Cobas, Meylan, France); CRP (Abbott, Chicago, USA); High-sensitive cardiac troponin I (Abbott); creatine kinase (Abbott) and creatinine (Abbott).

### 4.4. Statistical Analysis

#### 4.4.1. Biomarker Analysis

The quality of our paired dataset granted us to perform paired analysis. Unpaired analyses were run afterwards to corroborate the results. All statistical tests were generated by R Software, Version 3.1.1 (R Core Team, Vienna, Austria). The data are expressed as mean ± standard deviation or median with IQR for continuous variables and as frequency, with a percentage for categorical variables. We used the Shapiro-Wilk test to determine the parametric nature of the variables. The Mann-Whitney test was used for non-parametric variables and the Student’s test for parametric variables. For matching patients, comparisons of non-parametric quantitative variables were performed using the Wilcoxon signed-rank test and parametric variables with the Student’s *t*-test for matched data. Then, Benjamini correction was applied. A *p*-value ≤0.05 was considered statistically significant.

#### 4.4.2. Metabolomic Analysis

An unsupervised analysis was conducted using paired PCA to detect similar groups of patients and samples displaying an atypical metabolite profile (outliers). Paired PLS-DA was performed to discriminate between LVR and no-LVR patients based on their metabolic profile. PLS-based models are prone to overfitting, sometimes leading to overly optimistic models with poor predictive capabilities. To test the predictive capabilities of PLS-based models, data are often divided into training (~2/3 of samples) and test (~1/3 of samples) sets. Models are then constructed using the training set and their predictive capabilities are tested using the test set. A number of parameters are used to measure the performance of the models in the test set; however, the area under the Receiver Operating Characteristic is one of the most common parameters used because it is parsimonious, easy to interpret and enables performance comparison with other classification methods unrelated to PLS. Training set-based models with AUROC ≥ 0.8 on the test set are considered to be good predictive models whilst AUROC ≥ 0.9 is indicative of very good predictive models with a low degree of overfitting. Training and test set partition is usually done only once and randomly, based on the hypothesis that the probability of obtaining training and test sets representative of the entire population is much higher than that of obtaining training or test sets with extreme configurations that are totally different from the initial population. However, when dealing with small sample sizes, the probability of obtaining less representative training or test sets is far from negligible and could lead to misleading conclusions based on the training/test set partition strategy. Considering this issue and by allocating 22 samples (~2/3) to the training sets and 10 samples (~1/3) to the test sets in each group (LVR and no LVR), a total of 64,512,240 models are possible ($C_{22}^{32}$ = 64,512,240). To make calculations computationally feasible, a matrix with all these possible combinations was constructed and its columns, each representing a different combination, were sampled every 310 columns leading to 64,512,240/310 = 208,104 models. All of these models were run after unit variance scaling. For each model, the AUROC and its associated *p*-value for the test set were calculated. This *p*-value measures the probability, for a given AUROC, that the corresponding model was the mean model, randomly predicting patient status (remodeled or not) with no added information based on the metabolic profile. Since AUROCs and *p*-values cannot be considered as being normally distributed, median values instead of means were used to decide whether PLS models separate these two groups satisfactorily. A cut-off of 0.8 and 0.05 was selected for the median AUROC and median *p*-value, respectively. Global performance of pPLS on our data was considered as satisfactory only if median AUROC ≥ 0.8 and median *p*-value ≤ 0.05. In this case, variable selection was conducted based on the variable importance for the projection (VIP) and loading parameters. VIP values summarize the importance of each variable for the PLS-DA model, whereas the loading values are indicators of the relationship between the y vector containing the class information (i.e., glaucomatous or control) and variables in the X matrix (i.e., metabolites). Variables with a VIP value greater than unity are important for group discrimination. Multivariate analysis was performed using the R package mixOmics (http://www.Rproject.org).

## 5. Conclusions

The usual parameters that focused on infarction severity, such as infarct size and inflammatory biomarkers, differed between patients with and without LVR. The metabolomic approach performed soon after the onset of STEMI failed to discriminate between these patients.

## Figures and Tables

**Figure 1 ijms-20-00289-f001:**
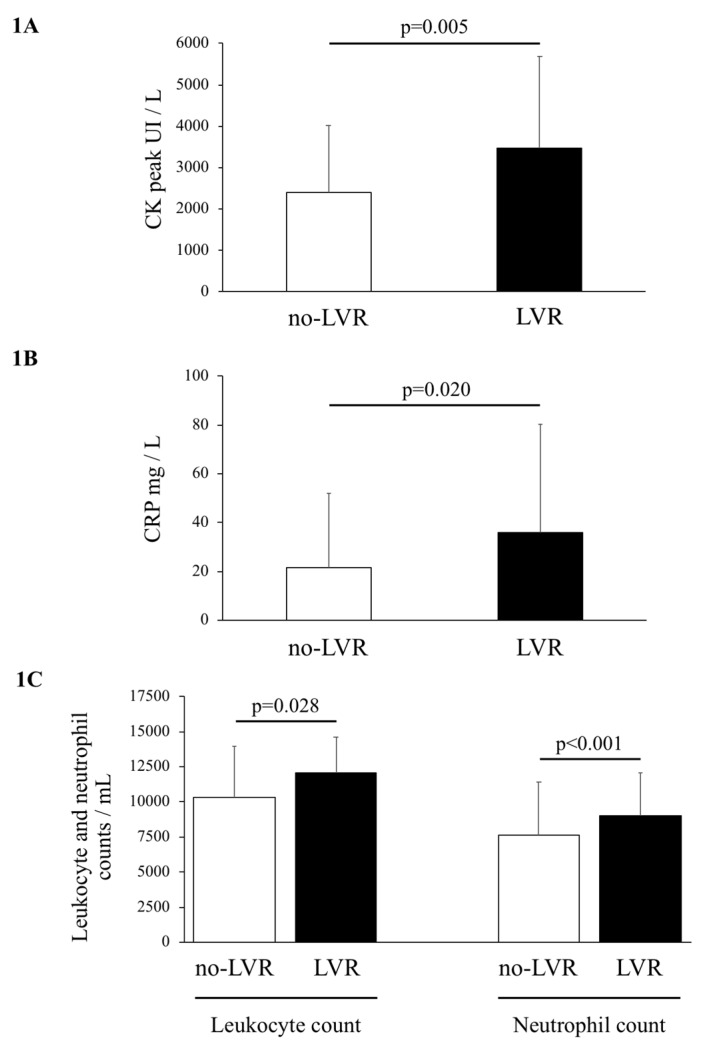
Creatine kinase peak (**A**), C-reactive protein (**B**), and leukocyte and neutrophil counts (**C**) among patients with and without left ventricular remodeling. LVR, left ventricular remodeling; CRP, C-reactive protein; CK, Creatine kinase. *p* Values to compare LVR patients with their controls.

**Figure 2 ijms-20-00289-f002:**
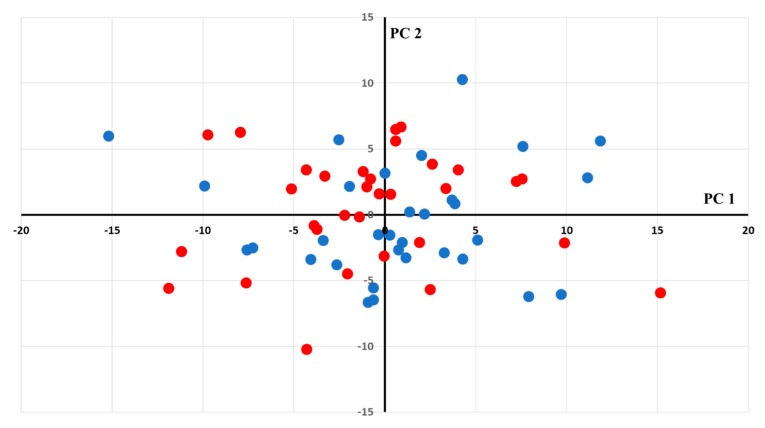
Paired principal component analysis (PCA) scatter plot obtained from the matrix of metabolites for the 32 patients affected by left ventricular remodeling (red circles) and their paired controls (blue circles). There is no clear grouping or outlier (according to Hotelling’s T^2^ range). PC1,2: Principal Components 1 and 2.

**Figure 3 ijms-20-00289-f003:**
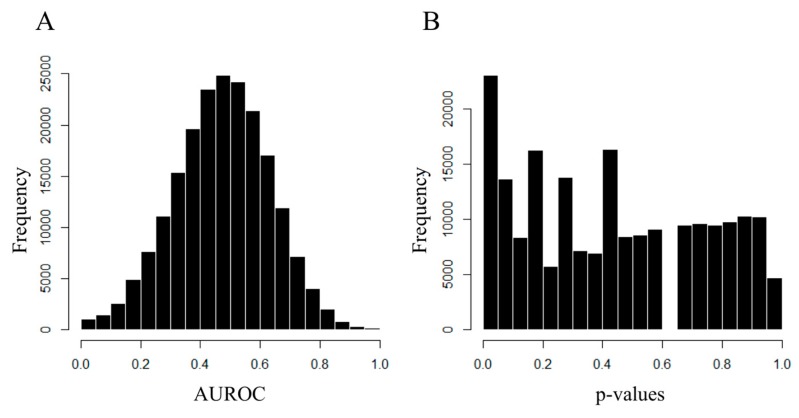
Metabolomics, area under the Receiver Operating Characteristic (AUROC) (**A**) and *p*-value (**B**) distribution for the 208,104 models tested. Each AUC and *p*-value evaluates the performance of a partial least square discriminant analysis (PLS-DA) model built with data from the training set. Distribution of the AUROC seems normal with mean and median equal to 0.48. *p*-value distribution seems abnormal and only 23,105 (11%) of these AUROCs has a *p*-value ≤ 0.05.

**Table 1 ijms-20-00289-t001:** The main characteristics of the study population.

Baseline Characteristics	Total *n* = 64	No-LVR *n* = 32	LVR *n* = 32	*p* Value
Anthropometry				
Male sex	52 (81%)	26 (81%)	26 (81%)	1.00
Age, years	58 ± 9	59 ± 8	58 ± 9	0.56
Body mass index, kg/m²	27 ± 4	27 ± 4	28 ± 3	0.17
Cardiovascular Risk Factors				
Hypertension	26 (41%)	8 (25%)	18 (56%)	0.010
Diabetes mellitus	9 (14%)	3 (9%)	6 (19%)	0.29
Hypercholesterolemia	34 (53%)	16 (50%)	18 (56%)	0.62
Current smoker	29 (45%)	12 (38%)	17 (53%)	0.22
STEMI Characteristics				
Anterior infarction	34 (53%)	17 (53%)	17 (53%)	1.00
Thrombectomy	31 (48%)	12 (38%)	19 (59%)	0.082
Complete revascularization	38 (59%)	19 (59%)	19 (59%)	1.00
Time to reperfusion (min)	293 ± 130	280 ± 117	307 ± 140	0.39
In-hospital heart failure	10 (16%)	2 (6%)	8 (25%)	0.041
Thrombolysis	14 (22%)	5 (16%)	9 (28%)	0.36
GPIIb/IIIa inhibitors	15 (23%)	8 (25%)	7 (22%)	0.77
Pharmaceutical Treatment at Discharge				
β-blockers	61 (95%)	31 (97%)	30 (94%)	0.56
ACE inhibitors	62 (97%)	32 (100%)	30 (94%)	0.16
Anti-aldosterone	25 (39%)	11 (34%)	14 (44%)	0.45

Baseline characteristics. ACE, angiotensin converting enzyme; LVR, Left ventricular remodeling; STEMI, ST-elevation myocardial infarction. *p* Values to compare LVR patients with their controls.

**Table 2 ijms-20-00289-t002:** Cardiac magnetic resonance (CMR) data.

CMR Data	Total *n* = 64	No-LVR *n* = 32	LVR *n* = 32	*p* Value
Baseline CMR				
LVEDV index, mL/m^2^	93.0 ± 18.9	94.7 ± 14.8	91.3 ± 22.4	0.48
LVESV index, mL/m^2^	48.2 ± 15.0	49.6 ± 12.2	46.8 ± 17.5	0.46
LVEF, %	48.2 ± 9.5	47.7 ± 9.0	48.6 ± 10.1	0.73
LV mass index, g/m^2^	59.2 ± 11.7	57.7 ± 8.5	60.7 ± 14.2	0.31
Infarct size, % LV	18.0 ± 12.5	15.8 ± 10.0	20.0 ± 14.4	0.19
3 Months CMR				
LVEDV index, mL/m^2^	94.1 ± 21.7	88.8 ± 16.2	99.3 ± 25.2	0.054
LVESV index, mL/m^2^	47.6 ± 19.0	42.1 ± 13.5	53.1 ± 22.1	0.019
LVEF, %	50.9 ± 9.8	53.1 ± 8.6	48.6 ± 10.4	0.064
LV mass index, g/m^2^	54.4 ± 10.5	53.2 ± 8.5	54.6 ± 12.2	0.60
Infarct size, % LV	16.4 ± 11.2	14.2 ± 9.4	17.3 ± 12	0.25
1 Year CMR				
LVEDV index, mL/m^2^	96.4 ± 25.1	86.6 ± 17.9	106.2 ± 27.6	0.001
LVESV index, mL/m^2^	49.0 ± 21.8	40.9 ± 14.5	57.2 ± 25.0	0.002
LVEF, %	50.4 ± 9.8	53.5 ± 8.9	47.3 ± 9.9	0.011
LV mass index, g/m^2^	52.3 ± 11.5	50.0 ± 8.9	54.6 ± 13.4	0.11
Infarct size, % LV	14.8 ± 8.9	13.0 ± 7.7	15.6 ± 10.2	0.25

LVR, left ventricular remodeling; CMR, cardiac magnetic resonance; LV, left ventricular; LVEDV, left ventricular end-diastolic volume; LVEF, left ventricular ejection fraction; LVESV, left ventricular end-systolic volume. *p* Values to compare LVR patients with their controls.

**Table 3 ijms-20-00289-t003:** Biomarker analysis at 48 h (mean ± standard deviation) after admission.

Biomarkers	Total *n* = 64	No-LVR *n* = 32	LVR *n* = 32	*p* Value
NT-proBNP, ng/L	1610 ± 1762	1375 ± 1630	1844 ± 1882	0.23
High-sensitive cardiac troponin I, ng/L	59,669 ± 80,127	43,557 ± 46,803	75,782 ± 101,594	0.26
Creatine kinase, UI/L	708 ± 791	566 ± 596	850 ± 935	0.27
Creatine kinase peak, UI/L	2930 ± 1995	2394 ± 1615	3466 ± 2211	0.005
CRP, mg/L	28.8 ± 38.3	21.7 ± 30.4	35.9 ± 44.3	0.020
Leukocyte count, /mL	11,187 ± 3240	10,346 ± 3626	12,028 ± 2593	0.028
Neutrophil count, /mL	8341 ± 3482	7596 ± 3822	9035 ± 3036	<0.001
Creatinine, µmol/L	73.5 ± 14.8	72.2 ± 15.3	74.8 ± 14.3	0.41

N-terminal prohormone of brain natriuretic peptide; CRP, C-reactive protein. *p* Values to compare LVR patients with their controls.

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
