# Peer review of "Metabolomic Approach in STEMI-Patients Undergoing Left Ventricular Remodeling"

_ijms, 2019, doi:10.3390/ijms20020289_

Round 1
Reviewer 1 Report
How the LVR was estimated in each patients? (Echo or MRI or both methods)
Could authors provide the dynamical changes in results of CMR analysis in total group and in both subgroups?
How 32 no-LVR patients were selected from total group of 192 patients ?
Why authors estimated biological markers only in one point - in 48 hours after the admission in a hospital but did not in other points?
Author Response
We are grateful to the reviewer for sharing his valuable comments and suggestions. We believe the requests improved substantially the quality of our manuscript.
[1. How the LVR was estimated in each patients? (Echo or MRI or both methods)]
Response: LVR was only defined by the means of MRI. Due to its low inter- and intra-observer variability, MRI is the best method to measure LV volumes, so that our cohort was systematically and on purpose followed by MRI. In order to improve our paper’s readability, we modified the first paragraph in the results section (page 2 lines 48-53).
[2. Could authors provide the dynamical changes in results of CMR analysis in total group and in both subgroups?]
Response: We now provide the changes from baseline to 12-month assessment in the manuscript (page 2 lines 54-55). “Mean change in LV end-systolic volume over time was -17.2 ±1.3% among no LVR patients, and +22.2 ±1.6% among LVR patients.”
[3. How 32 no-LVR patients were selected from total group of 192 patients ?]
Response: The selection process was as follows: we identified 32 LVR patients from our total cohort of 192 patients. LVR was defined by a change in left ventricular end-systolic volume ≥ 10% during follow-up. Then, out of the 160 remaining no-LVR patients, 32 were matched with regard to age, sex, left ventricular ejection fraction, infarct size, culprit artery and time to reperfusion. This is specified in the method section (page 7 lines 200-201), and we agree to provide this information earlier in the manuscript if requested by the reviewer or the editors.
[4. Why authors estimated biological markers only in one point - in 48 hours after the admission in a hospital but did not in other points?]
Response: We thank the reviewer for his comment. The purpose of our study was to identify new early biomarkers of LVR with an ulterior clinical motive to enhance therapeutic management. For now, there is no clear timepoint to evaluate biomarkers associated with LVR after a STEMI, as timing used in available papers vary broadly, ranging from 24 hours to seven days (Fertin Am J Cardiol. 2012 Jul 15;110(2):277–83.).
More, LVR can occur very early, that is a few hours after STEMI. We thought it was not judicious to perform our analysis too late after 48 hours. Conversely, due to the multitude of signaling pathways that occur just after STEMI, it would have been probably difficult to obtain a metabolomic signature too early after STEMI. So, 48 hours appeared as a good compromise, used in many other studies (Hsu Int J Med Sci. 2017 Jan 15;14(1):75-85.); Mather Int J Cardiol. 2013 Jun 20;166(2):458-64.); Orn Eur Heart J. 2009 May;30(10):1180-6.).
Hence, we are conscious that estimating biomarkers at 48 hours represent a limit of our study. This is now mentioned in the manuscript (page 7 lines 178-179).
Reviewer 2 Report
This is an interesting study exploring a novel approach to the Left ventricular remodeling (LVR) occurring after ST-segment elevation myocardial infarction (STEMI). Despite of the conclusion that the metabolomic approach did not discriminate patients with and without LVR, metabolomics analysis may yield novel predictive biomarkers that will potentially allow for an earlier medical intervention
As already stated by the authors, sample size may be one of the major limitation, affecting the result.
Recently, a study from Floegel A et al (Eur J Epidemiol. 2018 Jan;33(1):55-66. doi: 10.1007/s10654-017-0333-0) indicates that alterations in sphingomyelin and phosphatidylcholine metabolism, and particularly metabolites of the arachidonic acid pathway are independently associated with risk of MI in healthy adults.
The comprehensive integration of various omics data is becoming essential to provide a novel means of understanding the underlying pathophysiological mechanisms of myocardial infarction
Primary suggestion would be to increase the sample size to reach (more probably) a statistical significance. Second, it is desirable to consider further metabolites more specific of myocardial activity
Author Response
We warmly thank the reviewer for his time spent on our work and his precious comments. The remarks are of great interest.
[Primary suggestion would be to increase the sample size to reach (more probably) a statistical significance.]
Indeed, it is likely we may reach statistical difference with a larger population. However, recent cohorts of post-STEMI patients reveal LVR to become a quite infrequent phenomenon, occurring in only 15% of STEMI patients (Reindl et al., Eur Heart J Acute Cardiovasc Care. 2018 Apr 1:2048872618770600).
[Second, it is desirable to consider further metabolites more specific of myocardial activity]
We agree with the reviewer about the interest to focus on metabolites that may be more specific of myocardial activity. The kit we used is able to analyze 188 out of 40000 known metabolites. As there is no actual data on metabolic signature of LVR, we thought reliable to use such kit that assess 6 different biochemical families corresponding to distinct metabolic pathways, such as some hexoses or amino acid known to play an important role during LVR. Still it would be interesting to use another targeted analysis with different kit or an un-targeted analysis with a greater number of metabolites.
Reviewer 3 Report
The present manuscript reports the failed attempt to identify a metabolomic signature of LVR occurring after STEMI in a group of 32 patients and 32 controls.
The controls are selected by applying good inclusion criteria.
Nevertheless, the statistical analysis of the data was seriously flawed by the use of a paired approach.
Pairing indeed is useful to reveal differences between samples of the same subject before or after intervention or to monitor in time the disease progression in the same subject. Its use to compare different unrelated subjects (although sharing some common demographic or clinical features) is completely unjustified.
The lack of any metabolomic signature of LVR in the serum might be caused by the wrong statistical analysis.
The study should be repeated by applying unpaired multivariate and univariate analysis.
Author Response
Dear reviewer,
We are very grateful for the time spent on our work and the opportunity to discuss this methodological issue.
We completely agree with the fact that paired statistical methods are primarily suited for the analysis of several measures made on one single subject. However, paired tests can also be used even when the experimental design does not refer to longitudinal data. Paired analyses can also be performed for instance when one or more controls have been matched according to some relevant variables which one expects to influence the outcome (see, for example an extract from Woodward’s Epidemiology: study design and data analysis. 3rd edition” below*). Gender and age are commonly accepted as the minimum set of variables used for matching cases and controls. In our study, we used gender, age, left ventricular ejection fraction, infarct size, the culprit artery and time to reperfusion for matching every single case of LVR with a no-LVR control. This close pairing process was allowed by the large number of no-LVR patients. Hence, we considered the pairing to be of valuable quality in the matter of LV remodeling, so that we thought the use of univariate and multivariate statistical methods developed for paired samples to be granted.
Nevertheless, the comment of the reviewer was also raised by the methodologist of our Research team. Therefore, unpaired analyses were performed, including univariate (Student’s t-test on log-transformed data), multivariate (principal component analysis (PCA) and orthogonal partial least square-discriminant analyses (OPLS-DA)) considering control and case samples as independent. The results were the same compared to those obtained with methods developed for paired samples. Indeed, only two P-values (sphingomyelin 20:2, P-value=0.00925 and lysophosphatidylcholine 17:0, P-value=0.01846) were inferior to 0.05 but these P-values were greater than the corrected thresholds using Benjamini-Hochberg correction (0.0010 and 0.0016, respectively) used here to reduce risk I error inflation. Multivariate analysis showed no separation at all between samples from LVR and no-LVR patients (see Figure 1 of the present rebuttal letter) and the OPLS-DA algorithm failed building a predictive model using SIMCA 14.1 software.
>These latter results are now mentioned in the manuscript (page 5 line 115-116).
*Extract from Woodward, M. Epidemiology: study design and data analysis. 3rd edition(from section 2.7.4, p. 64)
Figure 1: Principal component analysis and orthogonal partial least square-discriminant analysis
(Samples considered as independent)
principal component analysis
orthogonal partial least square-discriminant analysis (opls-da)
No model
Round 2
Reviewer 3 Report
I appreciate the effort of repeating the statistical analyses.